# Optimizing Student-Driven Learning (SdL) through a Framework Designed for Tailoring Personal Student Paths

**Alberto Martinetti** 

Design, Production and Management Department, University of Twente, Drienerlolaan 5, 7522 NB Enschede, The Netherlands; a.martinetti@utwente.nl; Tel.: +31-(0)53-489-6609

**Abstract:** Our ever-changing and developing society constantly requires professions that did not exist 20 years ago. Students have to become professionals capable of steering their own career development and controlling their own learning process, at university and in their future profession. In order to reach these goals, lecturers have to understand the different needs of students in terms of knowledge and interests. This research offers a framework to help students deal with possible knowledge gaps and account for personal interests to match defined learning goals, utilizing the author's master's course in design for maintenance operations (DfMO) at the University of Twente as a basis for validation. First, a literature review was conducted on successful modern techniques of student-driven learning (SdL) to identify the best practices to use and possible pitfalls to avoid. Second, an analysis of the target group was carried out. Third, the research identified the most effective way to create such a tool (framework), taking into account the possible entry points of students. In particular, the research tried to understand to what extent it is possible and valuable to offer a student-driven approach. Finally, the tool was evaluated by representatives of the target group.

**Keywords:** student-driven learning; personalized learning; framework; engineering education

## 1. Introduction

Engineering is a broad multidisciplinary field. This is reflected in the increased variety of students in a single academic course in terms of foreknowledge and interests and skills.

Students with diverse backgrounds and even different knowledge levels are often present together in a similar learning environment.

This multidisciplinary characteristic addresses the needs of the industry: multidisciplinary engineers are needed to solve challenges the industry faces currently and will face in the future, ranging from design to end-of-life concerns, and from highly technical to more managerial problems. This scenario is not exclusive to a specific field, but a general trend in engineering [1]. The study takes as a case study the field of maintenance engineering due to the great variety of subjects students need to learn.

The maintenance engineering and operations (MEO) master track, offered by the faculty of engineering technology (ET) of the University of Twente and supported by the Twente Is Maintenance Excellence (TIME) consortium, is a highly multidisciplinary course. It does not belong to a single department, as other courses do, but crosses borders between departments in the faculty and even between faculties: the behavioral, management, and social science (BMS) faculty and the industrial engineering and business information systems (IEBIS) research group also contribute to the course. Future maintenance engineers need to be acquainted with highly technical topics and with managerial and design disciplines.

Therefore, the influx of students exhibits diverse backgrounds. Students who decide to join this specific master's course follow master's programs in mechanical engineering (ME), industrial design engineering (IDE), or industrial engineering (IE); they could have obtained a BSc degree in one of those, or in another topic, such as advanced technology (AT) or electrical engineering (EE). They decide to enter into the MEO course either because they chose it, or because they chose elective courses in the specialization. In addition, an increased influx of post-master's students (PhD and professional doctorate in engineering, PDEng) is observed, which also requires flexibility for the exit knowledge level, as these students can opt for post-master's variants of the courses. New educational methods are deemed necessary to achieve tailored learning paths for all students.

The key challenge is then twofold:

- How to offer a flexible program that allows students to make choices based on their own interests (student-centered or student-driven learning [2,3]), yet also guides them through the process, ensuring that the intended learning outcomes (ILOs) of the courses are achieved.
- How to give them the opportunity to continuously test their competency to evaluate whether they are on track.

## 2. Literature Review on Student-Driven Learning (SdL)-Student-Centered Learning (ScL)

In the literature, student-driven learning (SdL) or student-centered learning (ScL) environments are portrayed as more adequate than teacher-centered learning environments [4]. A teacher-centered learning environment, which is similar to traditional instruction, is said to discourage students from adopting a deep approach to studying [5]. Gow and Kember [6] reported that an SdL environment is less likely to induce a surface approach. Consequently, many researchers claimed that a transition in curricular and instructional approaches is needed, from teacher-centered to SdL environments [7,8]. However, such a transition can only be successful when the main actors (teachers and students) understand and agree with the underpinnings of the SdL environment. In other words, a smooth transition requires mutual adaptation of students' and teachers' instructional conceptions [9].

As also pointed out by Bell [10], SdL is a facilitated approach to learning that helps students to adopt a thorough and deep method of studying [5] and enhances their educational experience [11,12]. Learners may have increased freedom and control by adding personal objectives, determine most of the objectives and content, and choose and organize supplementary activities and materials (e.g., guest lecturers) to enhance their learning. Lecturers and students are partners in the learning process.

In essence, SdL is an approach in which students construct and reconstruct knowledge dynamically and their growing engagement is supported by motivational, cognitive, and social aspects. The active learning characteristic is advantageous and it encourages students to connect information and concepts to real-world scenarios [13]. This approach is rooted in constructivist and self-determination theories [14]. Moreover, as well discussed in [15,16], the role of the teacher in SdL and ScL moves from mere archives of information to a facilitator of learning, encouraging students to experience new activities and topics to study [17]. It is important to mention that Salter [18] highlighted that the introduction of Web 2.0 technology is empowering the use of SdL, in combination with adaptive learning features [19] and with flipped-classroom to give to the students the opportunity to plan, develop and evaluate their own learning process, training their meta-cognitive strategies [20]. This aspect was also underlined by Serrano et al. [21], pinpointing that blended learning became extremely popular in recent decades, showing it to be an effective approach for accommodating an increasingly diverse student population in higher education and enriching the learning environment.

Therefore, the implementation of digital technologies in higher education institutions is crucial as it determines to a great extent the efficiency and effectiveness of the teaching systems which correlate to innovation.

However, it needs to be said that despite the fact that the educational community agrees that the "one size fits all" model of teaching and learning is now behind us, we face real uncertainty about what is ahead. Personalized learning offers a new direction, but what does this term actually mean [22]?

At the moment, ScL is a 'container' notion that has not been clearly operationalized and has led to disparate and local interpretations and implementation in educational practice [23]. To move towards a "smoothly" embedded SdL or ScL in the different study programmes, both reference points need to be set [24] and evidences on how to structure efficient (for students and teachers) solutions need to be provided [25].

## 3. Methods and Materials

Using as a test case the design for maintenance operations (DfMO) course of the MEO specialization in mechanical engineering at the University of Twente, this paper proposes a flexible and scalable framework for tailoring a path to a student's needs and interests.

### 3.1. Methods: Research Methodology

Excluding the identification of the research problem and the literature study on SdL-ScL, considered to be the starting and motivating reason for the present work and the necessary picture to draw on existing studies, the research consisted of three main parts (Figure 1). First, data were collected to identify and understand the background and current master's programs. Second, the design and ideation of the framework for tailoring student paths was undertaken, using information retrieved from the literature and the data collected from the student population. Third and finally, the designed framework was evaluated with the use of focus groups in order to test its accuracy and adjust it according to the feedback received.

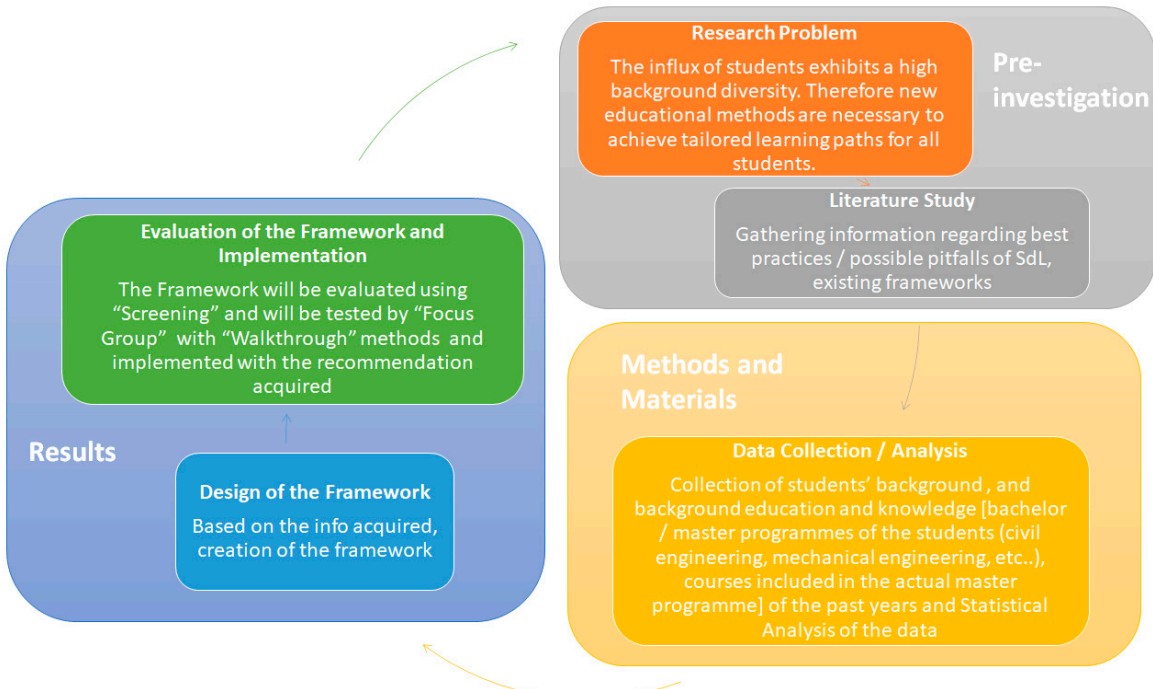

**Figure 1.** Main research steps inspired by the design-based approach.

The research adopted a design-based approach supported by qualitative evaluation. Following the matchboard method [3], the framework was then evaluated using members of the design research team (screening) and tested by representatives of the future target group (walkthrough). After the test, together with a representative group of former students, the next steps were discussed and proposed in order to refine the product.

*3.2. Materials: Data Collection and Analysis*

Data collection was performed in order to get a picture of the population of students (in terms of master's programs) who have joined the DfMO course over the past three editions.

Data collection is important in order to know the program directions of students who could benefit from the SdL method achieved with the framework. Moreover, this information helps in creating a tool that can guide them to clearly make their own path.

The data were acquired by analyzing the information available in learning management systems (LMS) such as Blackboard and Canvas of the course for the previous and current years.

The investigation of the current master's programs involved 18 students in academic year 2016–2017, 45 students in academic year 2017–2018, and 65 students in academic year 2018–2019. As Figure 2 shows, the variety of students has increased over the years. Even though a majority of students still come from mechanical engineering (ME) and industrial design engineering (IDE), it is important to realize that other master's programs are represented in DfMO, such as civil engineering (CE), industrial engineering and management (IEM), and computer science (CS). As pointed out in the Introduction, not only students with different competencies are present, but also students who have already achieved a master's degree and are now on a postgraduate trajectory (PhD or PDEng).

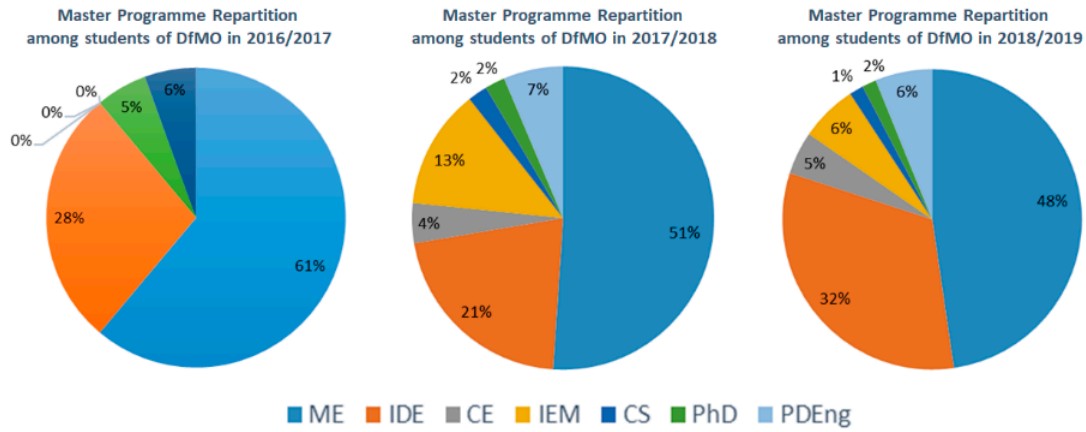

**Figure 2.** Master's program distribution in design for maintenance operations (DfMO) course from 2016 to 2019.

Nevertheless, it needs to be said that even in a situation where students have similar backgrounds, an SdL solution may improve their learning outcomes due to the different approaches and expectations.

## 4. Results

*4.1. Framework Design and Development*

As highlighted by Newman [26], "student-centered learning" is actually a complicated and "messy" idea that has encompassed a wide range of sometimes fundamentally different meanings, each holding important implications for education. Thus, educators need a more nuanced and more productive framework for conceptualizing student-centeredness in education. The first step of designing and developing the framework is to find relevant design criteria for creating such a model. To have some major guidelines to follow, open discussions with focus groups were conducted. Those and previous experiences gained with past master's courses pinpointed interesting features the framework should show:

- Understandability: how easy the framework is to understand;
- User-friendliness: how easily students can use the tool in order to choose their own educational path;

- Student profile: bachelor's/master's programs, courses included in the curricula;
- Interests: students should be able to drive the course assignments in a direction close to their field of study;
- Test results: self-diagnostic evaluations at the beginning of the course should help students understand which topic they should focus on in order to meet the learning goals.

The important aspects that were discovered drove the first design of the framework. Several design iterations were run before arriving at the last version (for the moment).

Figure 3 shows the main operators and symbols of the framework. The hexagon operator indicates the so-called body of knowledge: it represents competencies and skills already acquired or to be acquired by students. It can refer to either a generic or specific set of competencies. The yellow symbol in Figure 3 represents the student: it can be used to refer to students who have a general Bachelor of Science (BSc) in engineering or who are already following a specific master's program in engineering. The light red and blue symbol in Figure 3 indicates a test moment for students in order to evaluate the pre-knowledge or knowledge acquired during the course. Finally, the light red and blue rectangular symbol represents the meaning of gate tests.

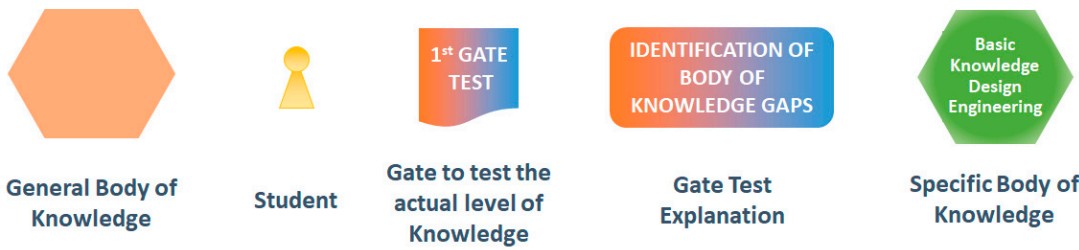

**Figure 3.** Main operators and symbols of the framework.

The framework starts with the body of knowledge acquired during the BSc in engineering (part 1). It is assumed that students with a BSc in engineering have already acquired enough skills without the need for pre-knowledge. It is important to mention that pre-knowledge in engineering was not tested during the evaluation phase. Therefore, students could immediately take the first gate test, if required, to enter part 2 of the framework. The first gate test was oriented to check whether students were lacking in the basic knowledge (maintenance engineering or design engineering) needed to start the DfMO course and pass it. The motivation to include an assessment in this specific moment was also the driving factor for starting the presented research. Students with different engineering backgrounds follow the course, and most of them are usually not familiar with either maintenance engineering or design engineering methods. Since the DfMO course needs both disciplines for successful completion, it is important that students realize which topics they should focus on in the very first part of the course in order to gain the necessary competency. The framework is designed to allow students the possibility to fill the knowledge gap in both topics, by receiving dedicated materials and extra support.

Once part 2 is completed (meaning the students have competency in maintenance engineering and design engineering), the body of knowledge referred to by the DfMO course is started. At this stage, students start acquiring competencies useful for achieving the ILOs of the course. It also means that every student (or group of students) starts working in parallel on an assignment tailored and designed by them and supported by the lecturer in order to apply DfMO concepts to specific fields related to the main lines of their master's program (or to their specific interests).

As pictured in Figure 4, to check whether they achieved the basic competencies of the DfMO course, students were asked to go through a second gate test. At this point of the course, according to Bloom's Taxonomy [27], first-level features such as remembering and understanding were assessed. The remaining competencies of the taxonomy (applying, analyzing, and evaluating) were evaluated through the tailored assignments. The second gate test and the assignment were finally introduced to

test the main competences of DfMO (indicated with a, b, and c), as they form the three main pillars of the course.

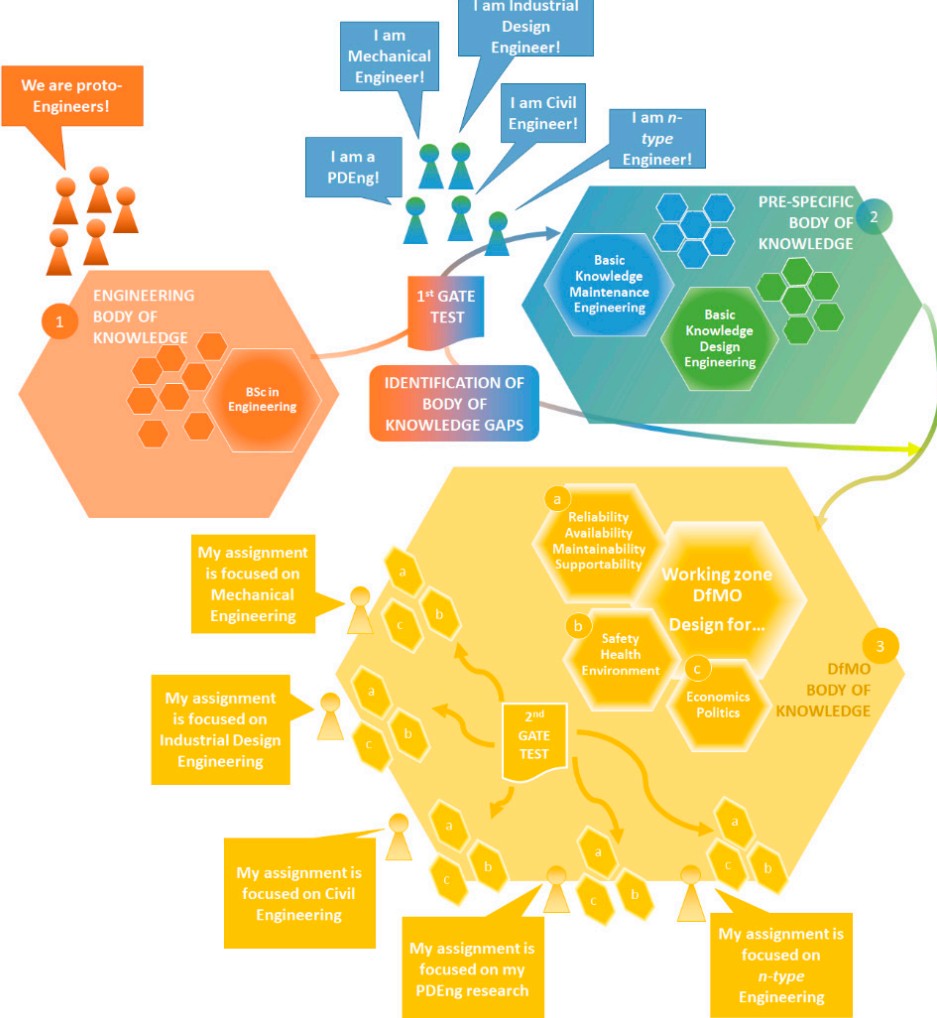

**Figure 4.** Framework based on student-directed learning (SdL) for the DfMO course.

*4.2. Framework Evaluation*

As anticipated, to analyze the efficiency of the created tool in improving students' engagement and learning process, an evaluation was carried out with the help of former students of the DfMO course.

Using the matchboard approach proposed in [3], students were asked to evaluate the framework based on the design requirements that arose during the design phase.

The evaluation looked for suggestions and remarks on the understandability and user-friendliness of the framework and representability of possible domains of interest.

The results of the evaluation are presented in Figure 5. A semi-quantitative method based on a Likert scale with a range of 0–5 was chosen to visualize the reflections of the students.

**Feedback distribution
of the evaluation of the Framework for DfMO course**

**Figure 5.** Evaluation of the framework.

The main points they were asked to reflect on were:

- How do you rate the understandability of the framework?
- How do you rate the opportunity to compensate for a lack of knowledge?
- How do you rate the possibility to drive the course to match your own interests?
- How do you rate self-diagnostic evaluations at the beginning and the middle of the course?

The small group of students reacted positively to the designed framework. As highlighted, the points related to understandability, the opportunity to compensate for a lack of knowledge, and the possibility to drive the course toward personal interests all scored around 90%. The point about self-diagnostics scored less than expected (67%). This means that there is still room for improvement. Investigating this low score in more detail with the students, it turned out that introducing a third gate could give a further advantage. The real benefit of adding this feature for students is not clear and needs to be further discussed.

## 5. Discussion and Conclusions

As highlighted in the introduction and also well discussed in [28], SdL is a powerful approach for increasing the student engagement and for allowing students with different background knowledge to work together. It is considered "the way to go" in terms of innovative teaching.

Nevertheless, it comes with issues and challenges, difficult sometimes to overcome due to the intrinsic nature. This motivation sparked the author to synthetize a framework that could act as a "high-level guide" for sketching SdL-based courses. Not an easy task.

Despite the complexity of the challenge, in general, the framework designed to guide students in using an SdL approach seems to be positively judged by the target group of students. The idea to check their actual level of pre-knowledge in advance and, based on that, create a personalized path to make up for existing gaps could bring benefits in terms of student performance. Moreover, it should increase the interest of the students in DfMO topics, letting them create their own assignments tailored to their master's program.

However, even if the framework was positively evaluated, it is far from being considered free of further refinements. The extremely small number of students involved in the test can only give an indication of the potential of the proposed approach. Before the framework is deployed in specific DfMO courses as a tool for lecturers and students, it will require several adjustments based on future

evaluations by students and educational experts. Future research should extensively evaluate the actual version of the framework with a large student sample in order to get feedback on the points highlighted in Figure 5. Having a statistically robust set of feedback data will provide a chance to express more critical and reliable judgments on the possible results achievable with the framework to create SdL paths.

**Funding:** The project was sponsored and supported by the Center for Engineering Education (CEE) of the 4TU Federation in the Netherlands and by the University of Twente. More information can be found to this link: https://www.4tu.nl/cee/en/innovation/community/10371/dr.-ir.-alberto-martinetti.

**Conflicts of Interest:** The author declares no conflict of interest.

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
