# Peer review of "Optimizing Student-Driven Learning (SdL) through a Framework Designed for Tailoring Personal Student Paths"

_education, doi:10.3390/educsci10090249_

Round 1
Reviewer 1 Report
The paper deals with an interesting issue on student-driven learning that research is needed to shed light on its conception, development and implementation. The paper includes important ideas but the whole structure and presentation does not allow to unfold these ideas. For example, the structure for presentation as suggested in the author’s guidelines is not followed-up. After the introduction, a section on research methodology is provided, followed-up by a section on student-driven learning and the followed-up by data collection/analysis etc.
For restructuring the paper, it is suggested that after the short introduction where the scope of the paper should be clearly presented, the author should proceed with a literature review on student-driven learning (section 3 in the paper), followed-up by the framework (section 5). Continue with research methodology, including section 2 and 4. Follow with a section about results, discussion and conclusion.
The literature on student-driven learning needs to be extended and analysed properly upon which the framework will be built on.
The author should be given a chance to restructure their paper, and be more analytic with every section.
Author Response
Dear Reviewer,
thank you very much for your valuable comments. I feel that they have helped me out a lot for improving the proper.
three main points came out from your review as 'conditio sine qua non' for publishing this work:
- a re-structure of the paper;
- a better literature research;
- english language revision;
- The paper has been re-structured following your suggestions: after the introduction, an improved literature research has been placed (discussed in point 2). Section 3 now is called "methods and materials", where the methodology of the study (3.1) and data collection (3.2) are been explained. After that, Section 4 "Results" has been placed. Here, the framework design and development (4.1) and the framework evaluation (4.2) are included. Finally, section 5 "Discussion and conclusions" wraps up the study pinpointing the next required validations steps.
- Literature research around Student-drive Learning and Student-centred Learning has been increase. 13 related references have been added (from 13-26 in the ref list).
- In addition to that, a complete language revision of the manuscript has been carried out by the professional experts offered by MDPI.
I want to thank you again for your valuable support.
Reviewer 2 Report
The manuscript is interesting. It provides a valuable framework for directing your own professional development and controlling your own learning process. However, it provides little information on how students are guided (by teachers) to follow their own learning paths. It would be interesting to provide such information.
On the other hand, I believe that the theoretical framework could be completed. The references offered on Student-driven Learning (SdL) are accurate but should be complemented with concrete references about self-directed learning.
Incorporte information about how de data were collected and analyzed.
Conclusions should be contrasted with other similar studies.
Author Response
Dear Reviewer,
thank you very much for your valuable comments. I feel that they have helped me out a lot for improving the proper.
Three main points came out from your review as 'conditio sine qua non' for publishing this work:
- a re-structure of the paper;
- a better literature research;
- English Language revision.
- The paper has been re-structured following your suggestions: after the introduction, an improved literature research has been placed (discussed in point 2). Section 3 now is called "methods and materials", where the methodology of the study (3.1) and data collection (3.2) are been explained. After that, Section 4 "Results" has been placed. Here, the framework design and development (4.1) and the framework evaluation (4.2) are included. Finally, section 5 "Discussion and conclusions" wraps up the study pinpointing the next required validations steps.
- Literature research around Student-drive Learning and Student-centred Learning has been increase. 13 related references have been added (from 13-26 in the ref list).
- In addition to that, a complete language revision of the manuscript has been carried out by the professional experts offered by MDPI.
I want to thank you again for your valuable support.
Round 2
Reviewer 1 Report
It has been improved but still the theoretical justification of student-driven learning needs more recent literature that is in accordance with the principles of student-driven paradigm. The same applies in the discussion of the results, which need to be substantiated with relevant previous research.
Author Response
Dear reviewer,
thanks again for your comments and suggestions.
Regarding the point "It has been improved but still the theoretical justification of student-driven learning needs more recent literature that is in accordance with the principles of student-driven paradigm. The same applies in the discussion of the results, which need to be substantiated with relevant previous research.", I have included a new paragraph in the discussion/conclusion session where I try to clarify with better precision the importance and challenges of Student-driven Learning approach, referring also to a new paper pubblised on this topic.
I hope this revision could successfully address your comment.
Thanks again fort the time spent during the revision.
With kind regards
Reviewer 2 Report
The changes suggested by the reviewers have been made.
Author Response
No further corrections need to be done according to the comment of the reviewer.
However, the author added a paragraph in the discussion/conclusion section and a reference to address the comments of the first reviewer. The new version of the paper is attached.
Best regards